# Atomic Arrangements of Graphene-like ZnO

**DOI:** 10.3390/nano11071833

**Published:** 2021-07-14

**Authors:** Jong Chan Yoon, Zonghoon Lee, Gyeong Hee Ryu

**Affiliations:** 1Department of Materials Science and Engineering, Ulsan National Institute of Science and Technology (UNIST), Ulsan 44919, Korea; yjc1526@unist.ac.kr (J.C.Y.); zhlee@unist.ac.kr (Z.L.); 2Center for Multidimensional Carbon Materials, Institute for Basic Science (IBS), Ulsan 44919, Korea; 3School of Materials Science and Engineering, Gyeongsang National University, Jinju 52828, Korea

**Keywords:** graphene-like ZnO, atomic arrangement, merging, aberration-corrected TEM

## Abstract

ZnO, which can exist in various dimensions such as bulk, thin films, nanorods, and quantum dots, has interesting physical properties depending on its dimensional structures. When a typical bulk wurtzite ZnO structure is thinned to an atomic level, it is converted into a hexagonal ZnO layer such as layered graphene. In this study, we report the atomic arrangement and structural merging behavior of graphene-like ZnO nanosheets transferred onto a monolayer graphene using aberration-corrected TEM. In the region to which an electron beam is continuously irradiated, it is confirmed that there is a directional tendency, which is that small-patched ZnO flakes are not only merging but also forming atomic migration of Zn and O atoms. This study suggests atomic alignments and rearrangements of the graphene-like ZnO, which are not considered in the wurtzite ZnO structure. In addition, this study also presents a new perspective on the atomic behavior when a bulk crystal structure, which is not an original layered structure, is converted into an atomic-thick layered two-dimensional structure.

## 1. Introduction

Bulk zinc oxide (ZnO) has a range of crystalline structures, namely wurtzite and zinc-blende structures [1]. The wurtzite ZnO is the most thermodynamically stable and exhibits strong electronic properties. It is also used in the flexible fabrication of electronic devices [2,3,4,5] and can transform into a variety of nanostructures [6]. When thinned down to a ZnO with atomic layers, planer ZnO [7,8,9,10], which resembles a graphene structure, can also form by expanding lattice parameter by 1.6% (a = 3.303 Å) [11] rather than the wurtzite ZnO. Since bulk ZnO is not a layered material, there is a limit to obtain atomic-thick nanosheets by physical and chemical exfoliation methods, such as methods related to graphene [12,13] and transition metal dichalcogenides [13,14,15]. However, it has been reported that ZnO nanosheets and graphene-like ZnO (g-ZnO) [7,8,9,10,16], which have trigonal planar coordination and are composed of Zn and O atoms alternately, can be synthesized by various deposition methods [7,8,9] with electron beam irradiation and a hydrothermal synthesis [10].

With the decreasing thickness of metal oxide semiconductor materials, unique electrical, mechanical, chemical, and optical properties are introduced, e.g., ZnO monolayer have increased band gap of ~4.0 eV comparing to bulk ZnO, which has a band gap of 3.37 eV, due to strong quantum confinement effects and graphene-like structure [8], while sustaining its direct band gap nature [17,18]. In addition, the g-ZnO is chemically stable [16,19,20] and is expected to exhibit high mechanical strength which originates from decreasing probability of finding defects by decreasing their thickness. In the same manner, low bending stiffness of the nanoscale thickness also enables g-ZnO to be applicable to the flexible electronics. Based on the extraordinary nature and properties of g-ZnO, it is very promising for transparent electronics, ultraviolet (UV) light emitters, chemical sensors, piezoelectric devices, switching electronics applications and photoactive devices [7,21,22], and ZnO QDs has potential applications in nanoscale devices [23].

We report the atomic arrangements at the edge of the g-ZnO nanosheets using aberration-corrected transmission electron microscopy (ACTEM). We explain the atomic reconstructions when the sheets merge at the edge of the g-ZnO nano-flakes. To observe that in detail, we use the g-ZnO nanosheets synthesized using an adaptive ionic layer epitaxy (AILE) method [24,25] and a monolayer graphene sheet as supporting layers inside transmission electron microscope (TEM). These results explain the atomic behavior of two-dimensional (2D) metal oxide semiconductors and direct observation of atomic arrangements from before to after merging.

## 2. Materials and Methods

### 2.1. Preparation of the Specimen

CVD-synthesized monolayer graphene film on 35 μm thick copper foil was purchased from Graphene Square (Graphene Square Inc., Seoul, Korea). Firstly, the graphene on copper foil was coated with ~200 nm poly methyl methacrylate (PMMA). Secondly, to release the graphene from the copper foil, the graphene on copper foil was floated onto copper etchant (Sigma Aldrich, St. Louis, MO, USA), then PMMA-coated graphene was transferred onto distilled water more than 3 times to remove the Cu etchant. The PMMA-coated graphene was then scooped up with TEM grids and annealed at 120 °C for 5 min. to firmly attach the PMMA-coated graphene onto the TEM grids. Lastly, the PMMA was dissolved with acetone for 1 day and the graphene on TEM grids were immersed into isopropyl alcohol several times to remove the acetone residue on the surface of the graphene. Synthesized ZnO nanosheets using the ILE [24,25] method were then scooped using the graphene-transferred TEM grid.

### 2.2. ARTEM Observations and STEM-EELS Spectra

Specimens were analyzed using an aberration-corrected FEI Titan Cubed TEM (FEI Titan3 G2 60–300), which was operated at an 80 kV acceleration voltage with a monochromator. The microscope provided a sub-Angstrom resolution at 80 kV and −13 ± 0.5 μm of spherical aberration (Cs). Typical electron beam densities were adjusted to ~6 × 105 e^−^nm^−2^. The atomic images were taken using a white atom contrast to obtain the actual atom positions under the properly focused conditions needed for direct image interpretation. STEM HAADF images and EELS spectra were recorded with a monochromatic beam at 80 kV with a probe size of 1.5 nm and an energy resolution of 0.8 eV, as measured from the full-width-at-half-maximum of the zero-loss peak.

## 3. Results and Discussion

We used synthesized g-ZnO nanosheets by an adaptive ionic layer epitaxy (AILE) method [24,25]. First, a monolayer graphene sheet is transferred on a TEM grid and then, the g-ZnO sheet is transferred on the monolayer graphene sheet (Figure 1a). We focused on the atomic arrangement and merging dynamics occurring from edge configurations of the ZnO nanosheets and nano-flakes. According to previous calculation and experimental research [7,8,9,16,17], wurtzite ZnO films can transform into a graphene-like structure, which is chemically stable. The wurtzite ZnO is schematically explained in Figure 1b. The Zn and O atoms form a tetrahedral configuration. G-ZnO has a hexagonal unit cell resembling a top-down view of the wurtzite ZnO structure in Figure 1b [26,27]. Since sp2 bonding of the hexagonal graphene-like structure is stronger than the sp3 bonding in the wurtzite structure, the Zn–O bond length of the g-ZnO structure is shorter as well [19,20,21].

For understanding the atomic movements of the g-ZnO nanosheets, we used the monolayer graphene sheets as a substrate layer because the ZnO sheets were not fully formed as lateral planar sheets. Since the lattice parameter of the ZnO and diameter of Zn atom is larger than their graphene, atomic dynamics of the ZnO sheets can be observed on the graphene sheet. An atomic image shows the g-ZnO sheets on the monolayer graphene sheets (as shown in Figure 1c) with an inset of fast Fourier transformation (FFT) showing spots of the graphene and spots of the ZnO sheets. Figure 1d,e shows a scanning transmission electron microscope high angle annular dark field (STEM-HAADF) image of a region where an electron energy loss (EEL) spectrum acquired and the EEL spectrum of the g-ZnO sheets on the graphene sheet, which confirms the presence of both Zn and O. The O-K edge where the peaks are located around 532 eV corresponds to oxygen atoms bonded to Zn in the ZnO nanosheet [28]. An inset of Figure 1d shows the C-K edge with π* and σ* peaks, which confirms the existence of graphene behind the ZnO sheets.

Figure 2 shows ACTEM images of edge configurations such as armchair (AC) and zigzag (ZZ) edge configurations nominally of the g-ZnO sheets. Previous research [8,9] shows that formation energies of O- and Zn-terminated ZZ edge configuration gradually decrease as the lateral size of the grown ZnO sheet increases at a lateral growth at the edge of the monolayer g-ZnO sheet. Furthermore, atomically extended ZZ edge configuration are observed (as shown in Figure 2a). For the AC edge configuration, although the lateral growth of ZnO is energetically favorable for the ZZ edge configurations rather that the AC edge configuration, the partial AC configuration is also able to be observed (as shown in Figure 2b).

In order to stabilize the edge of the synthetic g-ZnO sheet, but not the ZnO thinned from the bulky wurtzite ZnO, adatoms can be absorbed at the edge of the g-ZnO sheet. We observed that adatoms migrate along the edge of the ZnO sheet before being completely absorbed into the sheet (Figure 3). Although the migrating atoms at the edge might be carbons from carbon adsorbates and the graphene sheet, it is more likely that Zn atoms migrate because the Zn atom is clearly larger than the C. In addition, the positions in which the moving atoms settled are between O atoms designated at the O-terminated ZZ edge, which is confirmed using a line profile (as shown in Appendix A). Adatoms, colored with yellow dots, can attach and detach at the edge under continuous electron beam irradiation (Figure 3a,b). The attached adatom migrates freely along the O-terminated ZZ edge (Figure 3c–e) and finally separates from the ZZ edge (Figure 3f).

About the lateral growth of the g-ZnO sheet, there are two directions which are parallel and normal directions to the growth direction (as shown in Figure 4a). Since the ZZ edge configuration is more stable than the AC, the normal direction to the growth direction is preferable to that parallel to the growth direction [8]. However, when a merging occurs from the edges of g-ZnO nano-flakes (Figure 4b), adatoms bond parallel to the flakes (Figure 4c). The flake correspondingly expands in the normal direction gradually (Figure 4d–f), maintaining a well-aligned atomical state (as shown in an inset of Figure 4f). The merged flake can also merge with others (Figure 4g), which continuously merges and expands through the parallel and normal directions from the flake (Figure 4h,i). Notably, when flakes merge, ZnO adatoms are, firstly, bonded along to the parallel direction. The adatoms are then bonded along to the normal direction. The hole inside the flake formed during the continuous merging and expanding process (Figure 4f) is filled by continuous adsorption of the adatoms through the parallel and normal direction indicating green and cyan arrows (Figure 4j,k). Finally, a fully merged flake is formed (Figure 4l).

We observed, under electron beam irradiation, atomic arrangement of bridged-adatoms when merging at edges of the g-ZnO. Figure 5 shows successive ACTEM images at the edges. This area was chosen for monitoring for its atomic arrangements facilitating direct atomic dynamics. In Figure 5a, each edge atomic configuration is confirmed using intensity line profiles at the flakes (see Appendix A). Before merging, Zn and O atoms that are not completely bound to the ZnO sheet on the graphene sheet may randomly exist around the edges of the nano-flakes. When the continuous electron beam is irradiated, the ZnO flakes merge, exhibiting partial rectangular lattices, colored by a yellow, as an intermediate state (Figure 5b–e). Finally, the bridged atoms rearrange into hexagonal lattices (Figure 5f).

## 4. Conclusions

We report the atomic behavior of the g-ZnO nanosheets on the graphene sheet under electron beam irradiations. Previous theoretical and experimental works suggest interesting properties for these sheets, but our work focuses on atomic arrangements of the edge and merging between the ZnO nano-flakes. The g-ZnO has edge configurations (e.g., the ZZ and AC configurations) which were analogous to the graphene lattices. In addition, we observed the atomic migration by adatoms at the edge of the sheet. In terms of merging of the g-ZnO sheets, when the nano-flakes merge, they have the preferred directions depending on the growth direction. Once adatoms bond along the parallel direction the edge, the flakes expand toward the normal direction. Moreover, while the merging occurs, intermediate arrangements (e.g., rectangular lattices) form before the perfect merging into the hexagonal lattices. This work explains the atomic dynamics of the promising g-ZnO nanosheet, which suggests fundamental investigations of the 2D metal oxide materials.

## Figures and Tables

**Figure 1 nanomaterials-11-01833-f001:**
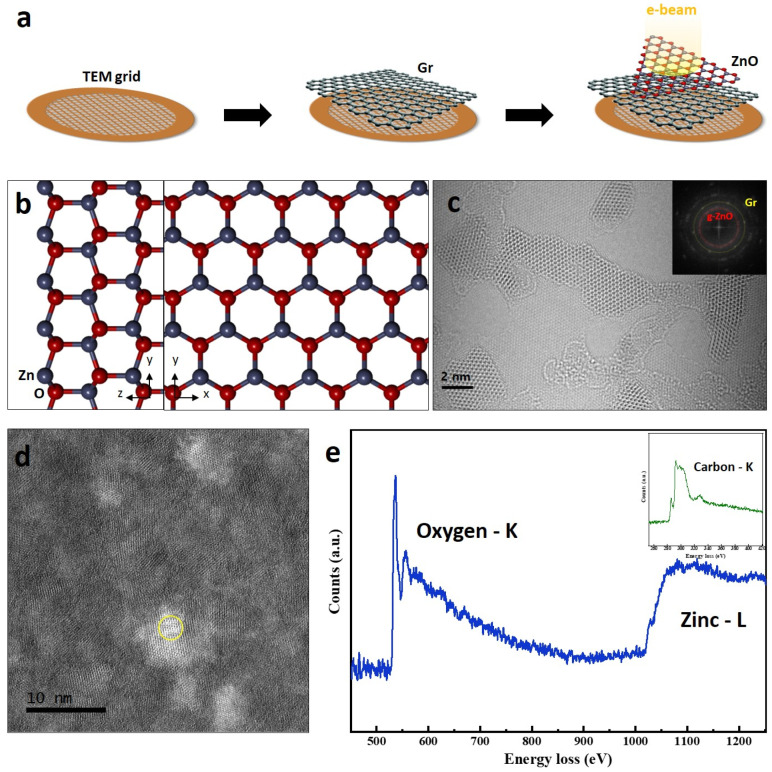
Graphene-like ZnO (g-ZnO) nanosheets on a monolayer graphene sheet. (**a**) Schematic showing a sample preparation for ACTEM imaging. (**b**) Atomic model showing top and side views of a wurtzite ZnO structure. (**c**) ACTEM image showing g-ZnO nanosheets on the monolayer graphene sheet with a FFT of the graphene and the ZnO sheets. (**d**) STEM HAADF image of the ZnO on the graphene. (**e**) EEL spectrum obtained from the yellow circle in (**a**) showing oxygen K-edge and zinc L-edge of the g-ZnO sheet on the graphene. The inset shows the carbon K-edge from the graphene.

**Figure 2 nanomaterials-11-01833-f002:**
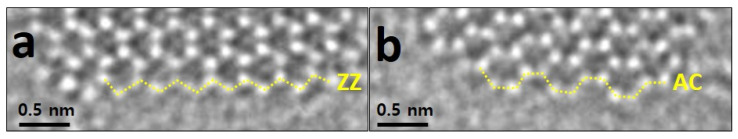
Edge configurations of g-ZnO. (**a**) ZZ edge configuration and (**b**) AC edge configuration.

**Figure 3 nanomaterials-11-01833-f003:**
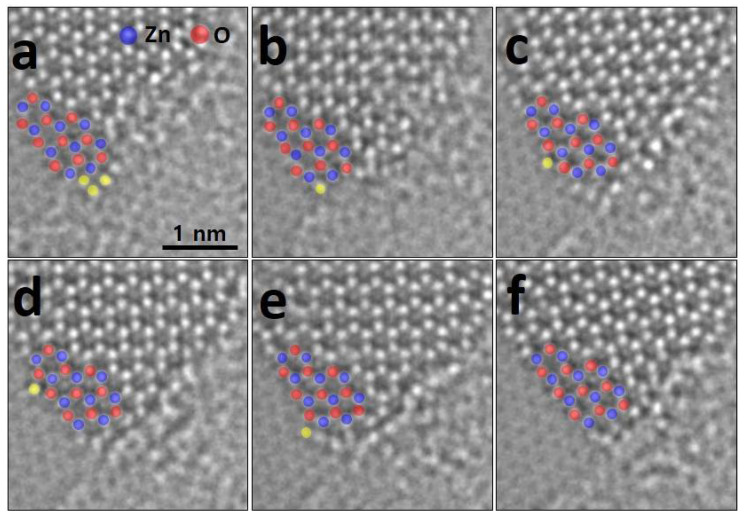
Atomic migration at the edge of the monolayer g-ZnO sheet. (**a**,**b**) successive images showing migrating adatoms, colored by yellow dots. (**c**–**e**) The adatom moves on top of dips in the ZZ configuration at the edge. (**f**) The adatom is finally detached.

**Figure 4 nanomaterials-11-01833-f004:**
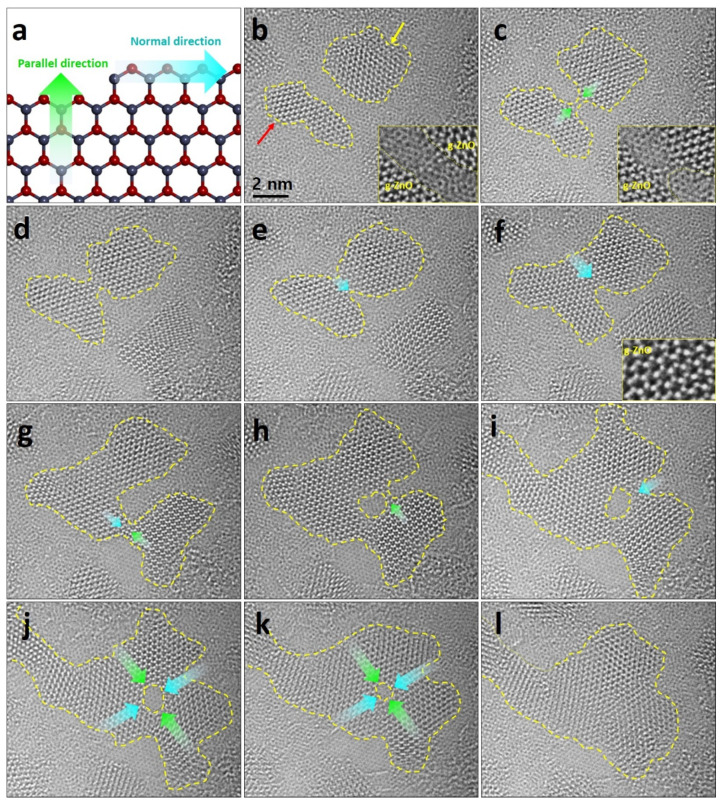
Merging of g-ZnO nano-flakes. (**a**) Atomic model showing normal and parallel to the growth direction. (**b**) Two graphene-like ZnO flakes before merging each other with magnified inset image showing edges of the flakes. Red and yellow arrows indicate that each flake is the same as flakes which are shown in Appendix A. (**c**) Initial state to merge the flakes along to the parallel directions of each flake. (**d**–**k**) Merging and expanding the flakes to the parallel and normal directions, indicated by green and cyan arrows. (**l**) Final g-ZnO flake after the process showing the behavior of merging and expanding.

**Figure 5 nanomaterials-11-01833-f005:**
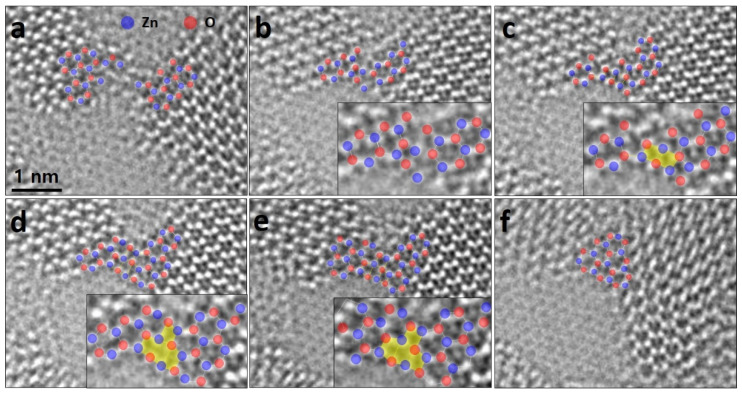
Atomic arrangement of bridged-atoms of the merging. (**a**) Initial state before the merging of the flakes. (**b**–**e**) Intermediate states showing the rectangular lattices with insets of the magnified bridged regions. (**f**) Final state after the merging. Raw images are supported as Appendix A.

## Data Availability

The data presented in this study are available in [insert article or Appendix A here].

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
