# Peer review of "Atomic Arrangements of Graphene-like ZnO"

_nanomaterials, 2021, doi:10.3390/nano11071833_

Round 1

Reviewer 1 Report

In this manuscript, the authors report on the structural modifications of atomic-thick 2D ZnO nanosheets under electron beam irradiation, studied by aberration-corrected TEM. This work provides insights into the atomic arrangements and adatom migration that occur, especially at the edges of the nanosheets, and thus provides important fundamental structural understanding for this type of materials. I recommend acceptance of the manuscript. I only have one comment for the authors: the yellow lines in figures 2 and 4 are difficult to see; the color should be changed to make it more clear.

Author Response

  • Following your suggestion, we have amended Figures 2 and 4.

Reviewer 2 Report

In the manuscript entitiled “Atomic Arrangements of Graphene-Like ZnO”, the authors studied the atomic arrangement and structural merging behavior of graphene-like ZnO nanosheets transferred onto a monolayer graphene using aberration-corrected TEM. The results are well presented well and scientifically sound. Therefore, I recommend that it should be published. There are also a few minor issues which should be addressed:

  1. In this the intruduction part, more details and discussions of the atomic thick ZnO nanosheets should be presented.

  1. The symbols and scale bars in the figures should be unified.

Author Response

  1. In this the introduction part, more details and discussions of the atomic thick ZnO nanosheets should be presented. 
  • Following your suggestion, we have added detailed discussions of the atomic thick ZnO nanosheets in the introduction part.
  1. The symbols and scale bars in the figures should be unified.
  • Following your suggestion, we have amended all.

Reviewer 3 Report

The authors investigate atomic arrangements at the edge of the g-ZnO nanosheets using aberration-corrected transmission electron microscopy (ACTEM). The atomic behavior of the g-ZnO nanosheets on the graphene sheet under electron beam irradiations were carried out and their possible mechanism were also dedeced. The work is interesting and would benefit the community of this research area. I have some suggestions for this draft before its can be published.

  1. In introduction part, authors can add more applications (refs.) for the g-ZnO in electronics applications.
  2. The size of texts, lines, and color in all figures can be improved for more clear presentation.

Author Response

  1. In introduction part, authors can add more applications (refs.) for the g-ZnO in electronics applications.
  • Following your suggestion, we have added detailed discussions of the atomic thick ZnO nanosheets in the introduction part.

  1. The size of texts, lines, and color in all figures can be improved for more clear presentation.
  • Following your suggestion, we have edited all figures.